# Evaluation of approaches to strengthen civil registration and vital statistics systems: A systematic review and synthesis of policies in 25 countries

Amitabh Bipin Suthar[1]*, Aleya Khalifa[1], Sherry Yin[1], Kristen Wenz[2], Doris Ma Fat[3], Samuel Lantei Mills[4], Erin Nichols[5], Carla AbouZahr[6], Srdjan Mrkic[7]

1 Center for Global Health, Centers for Disease Control and Prevention, Atlanta, Georgia, United States of America, 2 Programme Division, United Nations Children's Fund, New York City, New York, United States of America, 3 Health Statistics and Informatics Department, World Health Organization, Geneva, Switzerland, 4 Health, Nutrition, and Population Global Practice, World Bank Group, Washington DC, United States of America, 5 National Center for Health Statistics, Centers for Disease Control and Prevention, Hyattsville, Maryland, United States of America, 6 Bloomberg Data for Health Initiative, New York City, New York, United States of America, 7 Statistics Division, United Nations, New York City, New York, United States of America

* icf4@cdc.gov

**Data Availability Statement:** All relevant data are within the primary articles identified as eligible by this systematic review.

## Abstract

### Background

Civil registration and vital statistics (CRVS) systems play a key role in upholding human rights and generating data for health and good governance. They also can help monitor progress in achieving the United Nations Sustainable Development Goals. Although many countries have made substantial progress in strengthening their CRVS systems, most low- and middle-income countries still have underdeveloped systems. The objective of this systematic review is to identify national policies that can help countries strengthen their systems.

### Methods and findings

The ABI/INFORM, Embase, JSTOR, PubMed, and WHO Index Medicus databases were systematically searched for policies to improve birth and/or death registration on 24 January 2017. Global stakeholders were also contacted for relevant grey literature. For the purposes of this review, policies were categorised as supply, demand, incentive, penalty, or combination (i.e., at least two of the preceding policy approaches). Quantitative results on changes in vital event registration rates were presented for individual comparative articles. Qualitative systematic review methodology, including meta-ethnography, was used for qualitative syntheses on operational considerations encompassing acceptability to recipients and staff, human resource requirements, information technology or infrastructure requirements, costs to the health system, unintended effects, facilitators, and barriers. This study is registered with PROSPERO, number CRD42018085768. Thirty-five articles documenting experience in implementing policies to improve birth and/or death registration were identified. Although

**Funding:** This article has been supported in part by the US President's Emergency Plan for AIDS Relief through the Centers for Disease Control and Prevention under the terms of project number CGH2017233 (ABS). The funders had no role in study design, data collection and analysis, decision to publish, or preparation of the manuscript.

**Competing interests:** I have read the journal's policy and the authors of this manuscript have the following competing interests: ABS is a member of the Editorial Board of PLOS Medicine.

**Abbreviations:** CRVS, civil registration and vital statistics; ICD, International Classification of Diseases; ID, identification card; IDB, Inter-American Development Bank; IT, information technology; SDG, Sustainable Development Goal; WHO, World Health Organization.

25 countries representing all global regions (Africa, the Americas, Southeast Asia, the Western Pacific, Europe, and the Eastern Mediterranean) were reflected, there were limited countries from the Eastern Mediterranean and Europe regions. Twenty-four articles reported policy effects on birth and/or death registration. Twenty-one of the 24 articles found that the change in registration rate after the policy was positive, with two supply and one penalty articles being the exceptions. The qualitative syntheses identified 15 operational considerations across all policy categories. Human and financial resource requirements were not quantified. The primary limitation of this systematic review was the threat of publication bias wherein many countries may not have documented their experience; this threat is most concerning for policies that had neutral or negative effects.

## Conclusions

Our systematic review suggests that combination policy approaches, consisting of at least a supply and demand component, were consistently associated with improved registration rates in different geographical contexts. Operational considerations should be interpreted based on health system, governance, and sociocultural context. More evaluations and research are needed from the Eastern Mediterranean and Europe regions. Further research and evaluation are also needed to estimate the human and financial resource requirements required for different policies.

## Author summary

### Why was this study done?

- Civil registration and vital statistics (CRVS) systems generate foundational data for governance across sectors. Within public health, birth and death registrations are critical not only for empowering individuals' legal rights and for access to services but also in estimating health service needs, coverage, and impact.

- To our knowledge, no formal evidence review and synthesis has characterised which policies work and which do not for strengthening birth and death registration.

### What did the researchers do and find?

- We searched five literature databases and contacted global stakeholders to identify policies that improve birth and/or death registration and presented policy quantitative effects and qualitative syntheses.

- We identified 35 articles documenting experience in implementing policies to improve birth and/or death registration from 25 countries representing all global regions.

- Twenty-four articles reported supply, demand, incentive, penalty, or combination (i.e., at least two of the preceding policy approaches) effects on birth and/or death registration.

- Combination policy approaches, consisting of at least a supply and demand component, were consistently associated with improved registration rates across different geographical contexts.

## What do these findings mean?

- Countries interested in strengthening their CRVS systems may need to consider a combination of multiple policy approaches based on their health system, governance, and sociocultural context.

- Although this review and synthesis successfully identified many quantitative and qualitative data to strengthen CRVS systems, more policy evaluations and research are needed on effects from the Eastern Mediterranean and Europe regions and on human and financial resource requirements globally.

- Publication bias may have led investigators with neutral or negative findings to not document their findings; further research and evaluation is needed to understand which policies do not work.

## Introduction

Civil registration is defined as the continuous, permanent, compulsory, and universal recording of the occurrence and characteristics of vital events in accordance with the legal requirements in each nation [1]. Vital events captured in civil registration and vital statistics (CRVS) systems include the registration of births, deaths (including cause of death), marriages, adoptions, and divorces [1]. Civil registration and supporting legal documentation provide individuals with proof of legal identity, help establish their right to acquire nationality, allow individuals to exercise a broad range of rights, and facilitate access to essential services including social welfare, education, health, and legal protection [2]. Systematic compilation of civil registration data into vital statistics also provides the demographic information necessary for good governance [3]. For example, birth and death data can help monitor population growth and movement and inform fiscal policy. Within the health sector, functioning CRVS systems with a medically certified cause of death both provide an individual with the legal documents they need to access health, inheritance, and legal protection and the country with the data needed to estimate national and subnational burden of disease, the impact of different disease programmes, and the cost-effectiveness of disease interventions [4]. Birth and death registration data are also essential to inform health service needs and coverage. These functions are critical to monitoring progress in achieving the United Nations Sustainable Development Goals (SDGs) [5].

The systematic recording of vital events in many countries remains a serious challenge [5]. In the absence of reliable CRVS data, household surveys have become a key source of data to monitor levels and trends in births, deaths, and other core population indicators [6–9]. In most low- and middle-income countries, such surveys represent the sole source of this information. Unfortunately, many of these data sources are time limited, costly, externally supported, and not always current when published. Surveys also lack the local area CRVS data

most effective for local public health planning. Locally developed and sustainable CRVS systems can provide a legal identity from birth, the fundamental documentation to claim a nationality, and generate granular strategic information both to successfully deliver services and improve planning, budgeting, and programming for health and other sectors.

Public health authorities primarily focus on notifying births, deaths, and causes of deaths to the civil authority to enter into the registration system used for decision-making [10]. Global guidelines are useful in establishing CRVS norms and standards for countries. The United Nations Statistics Division provides comprehensive guidance on how CRVS systems can achieve universal coverage, continuity, confidentiality, data quality, and regular dissemination in order to be a dependable and primary source of vital statistics [11]. This guidance also suggests alternative sources and interim methods to generate vital statistics when CRVS systems are underdeveloped [11]. Technical guidance from the United Nations Children's Fund, World Health Organization (WHO), the World Bank Group, and various UN regional offices covers CRVS system strategic planning, legal frameworks, registration practices, birth certification, death certification and cause of death, quality of data according to the International Classification of Diseases (ICD), interim methods for vital statistics, and how to build political and community support for CRVS systems [10,12,13].

All countries agreed to achieve the SDGs that specify targets related to CRVS, including (1) by 2030, provide legal identity for all, including birth registration, (2) by 2020, enhance capacity-building support to developing countries to increase significantly the availability of high-quality, timely, and reliable disaggregated data, and (3) by 2030, build on existing initiatives to develop measurements of progress on sustainable development that complement gross domestic product, and support statistical capacity-building in developing countries [14]. CRVS systems also contribute to public health SDG targets, such as measuring progress towards ending the epidemics of HIV, tuberculosis, malaria, and neglected tropical diseases, reducing maternal and child deaths, and reducing deaths due to noncommunicable diseases and road traffic accidents [14,15]. Although there has been some progress in CRVS system development over the past two decades, birth and death registration rates continue to increase at a slow rate [16–18]. Worldwide, the proportion of children under five with a registered birth increased from 58% in 2000 to 65% in 2015 whilst the proportion of deaths registered increased from 36% to 38% during the same period [19]. In addition to informing national mortality and life expectancy trends, death registration is also critical because it is the first step before determining causes of death. Unfortunately, independent assessments indicate that the majority of registered deaths have issues surrounding the quality of cause-of-death ascertainment [20]. For example, the latest data available from WHO indicate that while an average of 48% of registered deaths included cause of death, 18% of ICD cause-of-death reports used ill-defined ICD codes [21]. Although significant progress was made in evaluating the role of information technology (IT) interventions in CRVS systems, non-technological interventions lack formal reviews and evaluations [22,23]. We systematically reviewed the evidence on national policy interventions to improve birth and death registration.

## Methods

### Study conduct

This systematic review was conducted in accordance with the PRISMA guidelines using a predefined protocol (International Prospective Register of Systematic Reviews identification number, CRD42018085768) [24,25] (S1 and S2 Texts). The ABI/INFORM, Embase, JSTOR, PubMed, and WHO Index Medicus databases were systematically searched without language, publication, or any other limits on 24 January 2017. Given that the Statistical Commission of

the United Nations adopted the *International Programme for Accelerating the Improvement of Vital Statistics and Civil Registration Systems* to assist countries with incomplete registration or entirely lacking a CRVS system in 1991, we included articles implemented and published from 1991 onward [26]. All sources cited in the 2007 and 2015 CRVS Lancet series were reviewed for inclusion [3,19,27–32]. Global stakeholders, including the Centers for Disease Control and Prevention's International Statistics Program, United Nations Children's Fund, UN Statistics Division, World Bank Group, and WHO, were also contacted to provide relevant publications on CRVS systems and policies.

### Eligibility criteria and search strategy

Per recommendations from the PRISMA group, eligibility criteria were based on key article characteristics: population, intervention, comparator, outcome, and design [24]. Specifically, sources were included when (1) they comprised a population eligible for birth and/or death registration, (2) the intervention was a new national policy (i.e., novel legislation or change in programme implementation designed to improve birth and/or death registration), (3) the comparator includes the lack of the new policy, (4) the outcome was birth registration rate, death registration rate, timeliness of birth registration, timeliness of death registration, and/or operational considerations (i.e., acceptability to persons registering births or deaths, acceptability to staff managing and implementing birth or death registration, human resource requirements, costs to the health system, adverse events, and/or facilitators or barriers learnt during implementation), and (5) the article design was a cross-sectional, cohort, case-control, or randomised controlled trial. Articles did not require a minimum time period of follow-up to be eligible. Articles describing operational considerations did not require a comparator arm to be included. The search strategies were designed by a librarian to identify articles that met these eligibility criteria (S3 Text).

### Study screening and extraction

Two investigators independently screened titles of all identified articles, followed by screening abstracts from relevant titles. The investigators then matched the full texts of all articles selected during abstract screening against the inclusion criteria. Disagreements were resolved through discussion with a third investigator. References for included articles were reviewed for additional reports. Articles failing to meet inclusion criteria were excluded from this review. Two investigators completed the data extraction using a standardised extraction form comprising four tables summarising setting, design, quantitative outcomes and results, and operational considerations.

### Quantitative and qualitative syntheses

For the purpose of establishing a policy framework for this systematic review, we used the following mechanisms and definitions to guide our categorisation of individual articles: (1) supply (policies focussed on increasing accessibility, acceptability, and/or affordability of registration services), (2) demand (policies focussed on increasing awareness for registration through information, education, communication, and/or advocacy), (3) penalty (policies that penalise citizens for failing to register a birth or death), (4) incentive (policies that encourage citizens to register a birth or death), and (5) combination (a combination of at least two of the preceding policy mechanisms). Given that numerators and denominators were not provided for most quantitative results, data stabilisation, meta-analyses, and heterogeneity assessment could not be performed [33–35]. Instead, we presented individual article results on changes in vital event registration rates during policy implementation. In cases in which multiple articles

reported quantitative results on an intervention from the same country, we reported results from the article with the longest period of follow-up.

We used Cochrane qualitative systematic review methodology, including meta-ethnography, to synthesise qualitative data [36–40]. Meta-ethnography involved reciprocal translational analysis (comparison), refutational synthesis (contrast), and line of argument synthesis (high-level synthesis) [41,42]. We developed themes based on quotations from each article. These themes were then categorised into one of our operational considerations: (1) acceptability to staff, (2) human resource requirements, (3) IT or infrastructure requirements, (4) costs to the health system, (5) unintended effects, (6) facilitators, and (7) barriers [43]. For reciprocal translation analysis, we compared similar themes from individual articles and synthesised operational considerations reflective of themes from multiple articles. For refutational synthesis, we contrasted themes and noted disagreements. Line of argument synthesis was used to synthesise operational considerations drawn from combination policy approaches [42].

## Results

### Search results

We identified 9,880 abstracts through database searches. After we removed duplicates and screened out nonrelevant abstracts, 450 full text articles were assessed for eligibility and 417 were excluded. We identified two additional articles from global stakeholders. In total, 35 articles, published from 1992 to 2016, met the eligibility criteria (Table 1; Fig 1; S1 Table) [44–77]. Articles that reported outcomes for multiple countries were disaggregated by country. In total, 25 countries were represented: 10 countries from Africa, 4 from the Americas, 5 from Southeast Asia, 2 from the Western Pacific, and 4 from Europe and the Eastern Mediterranean (Table 1). Nineteen articles from 15 countries reported experience implementing supply policies [44,47–53,56,59,61,66–68,74–76], one article from Mongolia reported on a demand policy [57], one article from Canada reported on a penalty [65], 5 articles from 5 countries reported experience implementing incentive policies [46,54,63,73,77], and 16 articles from 9 countries reported experience implementing a combination of different policies [12,45,55,58,60,62,64,69–73] (Table 1).

### Association with birth and death registration rates

Six combination policy articles reported results for birth registration, with all having a positive change in registration rates (Fig 2; S1 Table) [45,55,62,71,73]. One article from Tanzania reported that a supply policy had a negative change on the birth registration rate (Fig 2; S1 Table) [76]. One article reported that Canada's penalty policy had a negative change in birth registration [65] (Fig 2; S1 Table). One article from Zimbabwe reported that an incentive policy had a positive change in birth registration (Fig 2; S1 Table) [54]. Two combination policy articles reported results for death registration, with both positively affecting registration rates (Fig 3; S1 Table) [53,59]. Three supply policy articles reported results for death registration, with two positively affecting rates and one negatively affecting the rate (Fig 3; S1 Table) [48,50,76].

### Operational considerations

Across all policy categories, 15 operational considerations were meta-synthesised (Table 2) from eligible articles (S2 Table). This included nine considerations for supply-side policies, one for incentive policies, two for supply-side and demand-side policies, and three for supply, demand, and incentive combination policies.

**Table 1. Article characteristics, listed by country of implementation.**

| Article | Country (Region) | National birth registration rate | National death registration rate | Policy category | Policy description | Geographic scale | Time period | Vital events | Comparator used | Outcomes |
|---|---|---|---|---|---|---|---|---|---|---|
| Skiri, 2012 [74] | Albania (Europe) | 98.6 | 8 | Supply | Computerisation of the civil status registers in civil status offices. This consisted of software development, setting up the hardware, and training of local staff. | National | 2002–2004 | Births and deaths | N/A | Qualitative |
| UNICEF, 2010 [72] | Bangladesh (Southeast Asia) | 20.2 | Not reported | Supply, Demand, Incentive | Bangladesh implemented multilevel interventions to improve birth registration. In 2004, they legally required that individuals show a birth certificate for proof of age when accessing social services like school, marriage registration, and inheritance. They also decentralised the civil registration system by integrating it into Union Parishads, schools, and the immunisation programme. Finally, they raised awareness through campaigns and announcing a national registration day. | National | 1996 on | Births | Baseline year | Quantitative and qualitative |
| World Bank, 2015 [71] | Botswana (Africa) | 72.2 | Not reported | Supply, Demand, Incentive | Botswana introduced a multipronged strategy to improve death registration encompassing (1) cessation of all late registration fees for destitute persons, orphans, and vulnerable children, (2) relaxation of legal documentation for remote area dwellers, (3) decentralisation to district and subdistrict levels and introduction of on-site birth and death registration, (4) ad hoc campaign in Ghanzi, Maun, and Lethakeng districts, (5) information, education, and communication through electronic media, radio, and television, (6) birth certificate required for school enrollment and death certificate required for burial, and (7) national ID number provided at birth. | National | 2003 on | Births and deaths | N/A | Qualitative |
| Hunter, 2011 [62] | Brazil (Americas) | 95.9 | 93 | Incentive | In 2001, Brazil established the social welfare programme Bolsa Familia. It provides cash transfers to poor families conditional on school enrollment of their children. In 2003, it expanded to require basic healthcare practices as well. The programme requires a birth certificate of the child or children in order for the family to enroll. | National | 2001 on | Births | N/A | Qualitative |
| UNICEF, 2010 [72] | Brazil (Americas) | 95.9 | 93 | Supply, Demand, Incentive | The Brazilian government implemented various policy interventions to improve birth registration. After the legal reforms in 1997 to make birth certification free, changes to the civil registration system occurred from 2000 to 2011. In 1999, the Ministry of Health conducted an information campaign for birth registration. In 2000, the government created monetary incentives for civil registrars and health facilities to register births. This initially mandated states to create the monetary compensation themselves at state and local levels so that birth certificates would be issued without a fee. Later, the Ministry of Health issued the monetary incentives to hospitals directly for every birth registered. In addition, birth registration systems were integrated into maternity wards using outreach units. This project was accompanied by training of maternity ward staff. From 2004 to 2008, an online registration system was integrated with the maternity ward processes. | National | 1997–2008 | Births | Baseline year | Quantitative and qualitative |
| UNICEF, 2013 [70] | Brazil (Americas) | 95.9 | 93 | Supply, Incentive | Brazil introduced a multipronged strategy to improve birth registration encompassing (1) expansion of the network of civil registration in states with low registration rates through outreach units of notaries within maternity wards, (2) informatic linkage of civil registrars and maternity wards in one state, (3) USD 1.72 (2000 exchange rate) monetary incentive provided to hospitals for each child registered at maternity wards and provided with a birth certificate prior to discharge. | National | 1998–2008 | Births | Baseline year | Quantitative and qualitative |
| Woodward, 2003 [64] | Canada (Americas) | 100 | 100 | Penalty | In 1996, the Ontario government enacted a law that allowed municipalities to charge administrative fees for birth registration. | Ontario | 1996 | Births | Baseline year | Quantitative |
| Lu, 2002 [65] | China, Taiwan Province of China* (Western Pacific) | Not reported | 98 | Supply | Because most deaths occur at home in Taiwan, many doctors without a formal medical degree began to conduct home-based death certification as a full-time job and often certified over 100 deaths/year. | Taiwan | 1994 | Deaths | N/A | Qualitative |
| Silva, 2016 [46] | Ethiopia (Africa) | 6.6 | 65 | Supply | Use of community health workers for birth reporting and death reporting of children under 5 years of age. In Ethiopia, health extension workers were used who also provide preventive/curative healthcare and promotion. They were incentivised to also report vital events with a monthly transportation allowance (USD 12) and given backpacks and initial training. | Jimma, West Hararghe | January 2012–March 2013 | Births and deaths | N/A | Qualitative |

(Continued)

**Table 1.** (Continued)

| Article | Country (Region) | National birth registration rate | National death registration rate | Policy category | Policy description | Geographic scale | Time period | Vital events | Comparator used | Outcomes |
|---|---|---|---|---|---|---|---|---|---|---|
| Prata, 2012 [57] | Ethiopia (Africa) | 6.6 | Not reported | Supply, Demand | Community health workers, including community-based reproductive agents, birth attendants, and priests, were made responsible to report all births and deaths found in the community to the local health post. These workers were trained on how to educate and motivate families to report births and deaths that occur in the community. The registrations were then processed at the health post and recorded in government logbooks. These workers would locate new births or death through both household and clinic visits. | Tigray region | 2010–2011 | Births and deaths | N/A | Qualitative |
| UNICEF, 2010 [72] | The Gambia (Africa) | Not reported | Not reported | Supply, Demand | The Gambian government integrated birth registration processes into maternal and child health clinics starting in 2004, while decentralisation of registration duties began as early as 1996. All public health officers from district health facilities were required to register births and issue birth certificates. Many health facilities integrated birth registration with their immunisation services. Additionally, birth registration awareness campaigns have been used in conjunction with health promotion. In 2003, a campaign promoted and provided mobile birth registration along with the provision of treated mosquito nets. | National | 1996–2006 | Births | Baseline year | Quantitative |
| Fagernas, 2013 [54] | Ghana (Africa) | 70.5 | Not reported | Supply, Demand | The Ghanaian government developed and implemented a registration campaign from 2004 to 2005 to improve birth registration. Most of the campaign consisted of public education. Children were registered during community events using community registration volunteers. Birth registration was simultaneously strengthened in the health sector by incorporating it into child health promotion weeks and training healthcare workers. The campaign was accompanied by other changes, such as extending the deadline to register a birth before incurring a late registration fee. This time period was legally extended from 21 days to 1 year just before the campaign. | National | 2004–2005 | Births | Baseline year | Quantitative and qualitative |
| Ohemeng-Dapaah, 2010 [63] | Ghana (Africa) | 70.5 | Not reported | Supply, Demand | Civil registration was integrated with health facilities and electronic medical records systems. Community health workers also conducted active surveillance of vital events in the community and referred individuals to the health facility for registration. During this implementation, the health sector infrastructure and work force was scaled up to increase access to health and social services: additional health facilities were opened to decrease distance and the ratio of health staff to local population was greatly improved. | Bonsasso village | 2007–2009 | Births and deaths | N/A | Qualitative |
| Singh, 2012 [58] | India (Southeast Asia) | 71.9 | Not reported | Supply | In 2005, the responsibility of civil registration shifted from police stations to primary health centres (PHCs). At the PHCs, medical officers became registrars and pharmacists became sub-registrars. Nurse midwives and community health workers supported the registration process as well. | Haryana | 2005–2009 | Births and deaths | Baseline year | Quantitative and qualitative |
| Mony, 2011 [61] | India (Southeast Asia) | 71.9 | 8 | Supply, Demand | Civil registration was strengthened through multiple interventions: nongovernmental partnerships, sensitisation of professionals involved in registration, active surveillance of vital events, awareness campaigns, and public education. Lay informers who participated in active surveillance were paid a small fee if they reported a vital event within 3 days of the event. | Five subdistricts in southern India | September 2007–August 2008 | Births and deaths | Baseline year | Quantitative and qualitative |
| UNICEF, 2010 [72] | India (Southeast Asia) | 71.9 | 8 | Supply | In Delhi, the health department supported the computerisation of the birth registration process since 2003. Citizen Service Bureaus (CSB) were established in each zone of Delhi, which used the Online Institutional Registration (OLIR) system. Using this system, staff at hospitals or nursing homes register births and deaths online at the institution. The CSBs serve as a kiosk for individuals to access various identity-related services such as requesting a birth certificate. In 2006, Delhi also launched the project to link immunisation services with the birth registration process. Using maternity wards in Delhi, the registration and immunisation information systems were integrated. This way, healthcare workers at maternity wards provided both services together. This grew to include community-based services by midwives—which increased the coverage of the birth registration system. | Delhi | 2003 on | Births | N/A | Qualitative |
| WHO, 2013 [12] | India (Southeast Asia) | 71.9 | 8 | Supply, Demand | The intervention consisted of three main strategies: (1) awareness campaigns among the public to create demand for registration, (2) capacity building for officials responsible for registration, in turn improving access, and (3) stakeholder engagement to build support and hold the government accountable. | Four states of India | 2000 | Births | N/A | Qualitative |

(Continued)

**Table 1.** (Continued)

| Article | Country (Region) | National birth registration rate | National death registration rate | Policy category | Policy description | Geographic scale | Time period | Vital events | Comparator used | Outcomes |
|---|---|---|---|---|---|---|---|---|---|---|
| Modi, 2016 [43] | India (Southeast Asia) | 71.9 | 69 | Supply | Mobile-phone Technology for Community Health Operation (ImTeCHO) is an application that was implemented to help Accredited Social Health Activists (ASHAs) complete registration processes. This intervention streamlined registration processes for improved access to services. | Random sample of five villages in Gujarat | 2013 on | Births and deaths | N/A | Qualitative |
| Duff, 2016 [45] | Indonesia (Southeast Asia) | 68.5 | Not reported | Supply | 2013 legal amendment to eliminate fees for all civil registration documents. | National | 2013 | Births | N/A | Qualitative |
| Dababneh, 2015 [47] | Jordan (Eastern Mediterranean) | 99.1 | Not reported | Supply | The Ministry of Health (MOH) and partners implemented multifaceted improvements to the death notification system put forth by a five-point plan: establishing a cause-of-death coding unit, modifying the death notification form, training on the ICD and Related Health Problems, appointing focal points for supervision and quality control, and tabulation and reporting. | National | 2003 on | Deaths | Death notification forms received by MOH as a percentage of forms reported to the Civil Status and Passports Department | Quantitative |
| Toivanen, 2011 [60] | Liberia (Africa) | 24.6 | 53 | Supply | The Ministry of Health and Social Welfare (MOHSW) implemented mobile birth registration to computerise the registration process. A birth registration server is housed at MOHSW and receives data from text messages from mobile phones. Registrars (users of the software) were trained. | Bomi county | 2007–2010 | Births | N/A | Qualitative |
| Gadabu, 2014 [73] | Malawi (Africa) | 67.2 | 78 | Supply | An electronic register was developed and implemented for a rural village in Malawi. A low-power touch screen computer was provided to the village headmen. These headmen were trained on its use for 5 days. | Chalasa village, Traditional Authority Mtema, Lilongwe District | March 2013 | Births and deaths | N/A | Qualitative |
| Silva, 2016 [46] | Malawi (Africa) | 67.2 | Not reported | Supply | Use of community health workers for birth reporting and death reporting of children under 5 years of age. In Malawi, health surveillance assistants were used who also provide preventive/curative healthcare and promotion. These workers were incentivised with quarterly airtime for phone calls, data review meetings, and given a participation allowance of USD 11 per data review meeting (every 3 months). These workers were given a village health register, cell phone, and backpack. | Balaka, Salima | January 2010–December 2013 | Births and deaths | N/A | Qualitative |
| Singogo, 2013 [51] | Malawi (Africa) | 67.2 | Not reported | Supply | The Malawian government introduced paper-based village registers in 2007 in an effort to decentralise the civil registration system. All districts were using village registers by 2011. The village headpersons (VHs) are responsible for keeping and updating the registers with birth and death events that occur in their villages. | Traditional Authority Mwambo, Zomba District | 2007–2011 | Births and deaths | Maternity registers (for birth events only) | Quantitative and qualitative |
| Silva, 2016 [46] | Mali (Africa) | 84.3 | 91 | Supply | Use of community health workers for birth reporting and death reporting of children under 5 years of age. In Mali, lay volunteer health workers were used who also provide health promotion. These workers were incentivised with USD 10 per month and USD 2 for airtime per month. They received registers and food/transport to attend quarterly review meetings. | Barrouéli, Niono | July 2012–September 2013 | Births and deaths | N/A | Qualitative |
| Lhamsuren, 2012 [56] | Mongolia (Western Pacific) | 99.3 | 92 | Demand | The Mongolian government developed and implemented the strategy, Reaching Every District (RED). RED is package for both health and social services for the urban poor. Specifically, a nurse in each local area is tasked to walk house to house and request new clients to register any vital events, and direct them to the health facility or registration office for registration. Under the RED strategy, local areas were identified through an analysis of social vulnerability and were targeted with a package of services. | Socially vulnerable areas | 2009–2010 | Births and deaths | N/A | Qualitative |

(Continued)

**Table 1.** (Continued)

| Article | Country (Region) | National birth registration rate | National death registration rate | Policy category | Policy description | Geographic scale | Time period | Vital events | Comparator used | Outcomes |
|---|---|---|---|---|---|---|---|---|---|---|
| Prybylski, 1992 [67] | Papua New Guinea (Western Pacific) | Not reported | 98 | Supply | As a part of a multipurpose birth registration scheme, a new computerised health information system was established and a new birth certificate/registration form was introduced to clinics that provided antenatal, postpartum, and well-child care. The birth certificate was intended to promote supervised delivery because it was offered to the mother for free if she delivered in a health facility or under supervision of a midwife. Otherwise, a small fee has to be paid to obtain the birth certificate. | Southern Highlands Province | 1988 | Births | N/A | Qualitative |
| Curioso, 2013 [55] | Peru (Americas) | 96.7 | Not reported | Supply | Peru implemented a system that allowed for real-time online registration of birth events while the mother was in the delivery room. | National | 2012–2013 | Births | N/A | Qualitative |
| WHO, 2013 [12] | Peru (Americas) | 96.7 | 69 | Supply, Demand | The Peruvian government, led by the Ministry of Women and Social Development (MIMDES) and the Alliance for Citizens' Rights, developed and implemented an advocacy campaign to increase birth registration coverage. The campaign was designed from a community-based strategy and changed the legal framework that 'discriminated against unmarried women and their children'. For example, the new bill made it possible for single women to declare the name of the father (a required piece of information) during registration without the father present. The advocacy campaign delivered the message that 'every child has the right to two last names'. The campaign consisted of mass media activities over the course of 4 weeks. | National | 2004–2006 | Births | N/A | Qualitative |
| IDB, 2009 [68] | Peru (Americas) | 96.7 | 69 | Supply, Demand | Peru adopted the National Civil Registry and Identification Organization (RENIEC) in 1993 and has undergone changes in the late 1990s and early 2000s. This organisation is autonomous from the Peruvian government, which allows it to evolve in response to its own technical needs. RENIEC implements rigorous training programmes to professionalise staff working on civil registration and identification. It spawned its own training institute, the Center for Higher Education in Registration (CAER). RENIEC has worked to improve the infrastructure surrounding registration, such as registration offices, auxiliary registry offices in public hospitals, call centres, home services for handicapped people, and free identity document campaigns for underserved populations. RENIEC conducts mobile registration using internet-based services in rural parts of Peru. | National | 1993 on | Births and deaths | N/A | Qualitative |
| Garenne, 2016 [44] | Republic of South Africa (Africa) | 85 | 91 | Supply, Demand, Incentive | Combination of financial incentives, advocacy through campaigns and wide-scale information sharing, and reorganisation of CRVS to strongly involve health personnel. One major financial incentive was that a birth certificate is required for child support grants, which provides mothers with substantial financial support. | Agincourt (rural area) | 1992–2014 | Births and deaths | Baseline year | Quantitative and qualitative |
| Joubert, 2013 [52] | Republic of South Africa (Africa) | 85 | Not reported | Supply | Multifaceted initiatives to scale up CRVS took place after the passing of the Births and Deaths Registration Act of 1992 and the Interim Constitution of South Africa in 1993. These initiatives include a new death notification form, task teams in each province to assist in the adoption of the new form, new guidelines for birth and death registration, birth registration forms made available for mothers at their delivery, and training health workers to support mothers in how to submit the birth registration forms. | National | 1992 on | Deaths | Baseline year | Quantitative |
| WHO, 2013 [12] | Republic of South Africa (Africa) | 85 | 91 | Supply, Demand, Incentive | After Apartheid, the South African government used a mixture of interventions to improve birth and death registration. Using leadership, political commitment, advocacy campaigns, and governmental partnerships, they were able to make birth registration universal. Government agencies raised awareness about registration and conducted outreach among communities, including local village chiefs. Staff involved in registration underwent training. To increase access among hard-to-reach populations, the registration office used mobile facilities. Some government services were made to require a birth certificate, such as child support grants or school enrollment. | National | 1997–2004 | Births and deaths | N/A | Qualitative |
| Upham, 2012 [59] | Republic of South Africa (Africa) | 85 | 91 | Supply, Demand, Incentive | Department of Health established a National Health Information System. Department of Home Affairs introduced new registration forms, raised awareness, and performed outreach to communities and village chiefs. Birth certificates were required for school enrollment. Community interventions included working with village chiefs through the registration process, providing child support grants to registered births, and setting up mobile registration sites. | National | 1997–2004 | Births and deaths | Baseline year | Quantitative and qualitative |

(Continued)

Table 1. (Continued)

| Article | Country (Region) | National birth registration rate | National death registration rate | Policy category | Policy description | Geographic scale | Time period | Vital events | Comparator used | Outcomes |
|---|---|---|---|---|---|---|---|---|---|---|
| Rao, 2004 [69] | Republic of South Africa (Africa) | 85 | 91 | Supply, Demand | Multi-stakeholder provincial task teams facilitated the introduction of a new death notification form, distributed manuals and trained relevant personnel nationally, and developed strategies to improve registration. | National | 1990 on | Deaths | Baseline year | Quantitative |
| Kabengele, 2014 [50] | Switzerland (Europe) | 100 | 8 | Supply | Collecting mortality data from funeral homes instead of from the conventional Federal Statistics Office. | Canton of Geneva | 2005–2010 | Deaths | N/A | Qualitative |
| Kabadi, 2013 [75] | Tanzania (Africa) | 26.4 | Not reported | Supply | A computer application was developed to receive text messages from community civil registration officers when a death or birth occurred. The text would consist of all the information required on the notification form and would be sent to the district registrar's office. The Village Executive Officer accessed this application and viewed trends in notifications, and saw when a notification was made but no certificate was acquired. This helped the Village Executive Officer to follow up with specific families to complete the registration process. Staff were recruited and trained. | Rufiji District | September 2012–March 2013 | Births and deaths | Baseline period | Quantitative and qualitative |
| Tangcharoensathien, 2014 [76] | Thailand (Southeast Asia) | 99.4 | 84.9 | Incentive | Computerised civil registration system integrated with national unique ID (that has utility for a range of government services) and sharing and interoperability among member databases held by three insurance schemes. | National | 2001 on | Births and deaths | N/A | Qualitative |
| Ozdemir, 2015 [49] | Turkey (Europe) | 98.8 | 4 | Supply | The Turkish Statistical Institute Death Reporting System (TURKSTAT-DRS) has undergone various governmental reforms since 2009. For example, the TURKSTAT system was integrated with the Central Population Administrative System (MERNIS) reporting form. This was useful because many rural areas lacking health practitioners only report the cause of death with this simple one-line MERNIS form, and register the deaths with the village headman instead. These new reforms reconcile between the TURKSTAT and MERNIS system by implementing routine system procedures for interoperability. Additionally, there has been enhanced coordination between the various local agencies responsible for death registration—with the main aim of improving registration completeness. | National, Izmir | 2001–2013 | Deaths | Baseline period | Quantitative |
| Starr, 1995 [66] | United States (Americas) | 100 | Not reported | Supply | Many US states began adopting electronic birth certificates (EBCs) in the 1980s–1990s. Different registration areas implemented different EBC systems, and coverage varied from 1% to 100%. | 46 states; New York City; and Washington, DC | 1980–1994 | Births | N/A | Qualitative |
| Tripp, 2015 [48] | US (Americas) | 100 | 100 | Supply | The Utah State Office of Vital Records and Statistics implemented an electronic death registration system to facilitate swifter death certification and registration. The system is used by funeral directors, physicians, and medical examiners and health department officials. The system overall helps government registrars to spend less time and resources on registration because it eliminates paper-based systems. | Utah State | 2006 on | Deaths | N/A | Qualitative |
| Robertson, 2013 [53] | Zimbabwe (Africa) | 43.5 | Not reported | Incentive | Households selected at random were enrolled into a conditional cash transfer programme (CCT), an unconditional cash transfer programme (UCT), or neither. In the UCT group, households received USD 18 and USD 4 per child every 2 months. In the CCT group, households received the cash transfer if they had complied with a set of conditions related to social welfare. One such condition is that the family had to have applied for a birth certificate within 3 months for any children in the family not yet registered. If the household complied with all the conditions, they received a card that they could bring to a pay point every 2 months to receive their cash transfer. | Manicaland | 2010–2011 | Births | Control group | Quantitative |

*Taiwan has not held a seat in the United Nations since China's rights were restored in 1971 by Resolution 2758/26. Moreover, the 1979 US-PRC Joint Communique switched US diplomatic recognition from Taipei to Beijing and recognised the Government of the People's Republic of China as the sole legal government of China. The United Nations Population Division reports estimates for this region as 'China, Taiwan Province of China'.

Abbreviations: CRVS, civil registration and vital statistics; ICD, International Classification of Diseases; ID, identification card; IDB, Inter-American Development Bank; N/A, not applicable; UNICEF, United Nations Children's Fund; USD, United States dollar

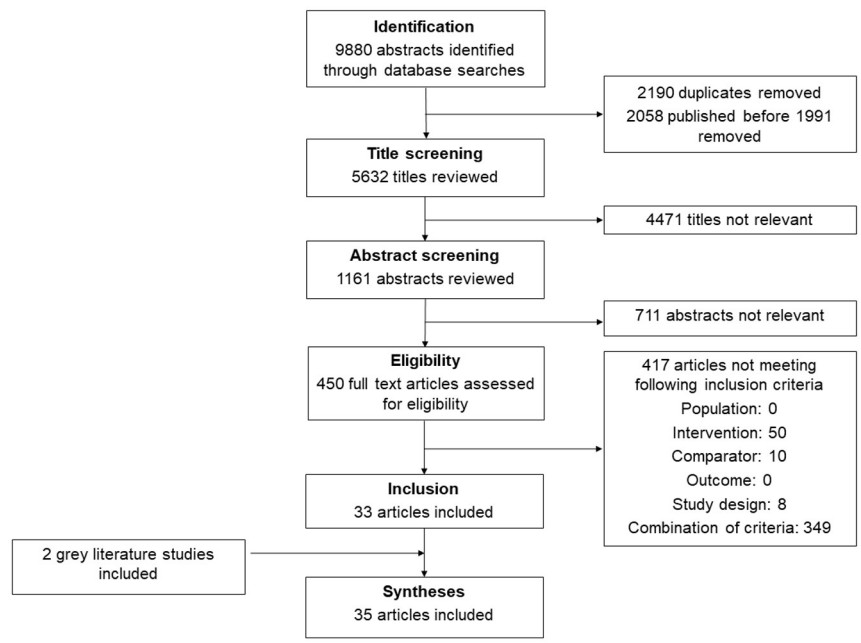

**Fig 1. Flow of information during different phases of the systematic review.**

## Discussion

This systematic review indicates that new policies have the potential to have positive long-term effects on improving birth and death registration rates. However, there are a variety of considerations needed to interpret these data correctly. For example, an intervention implemented at

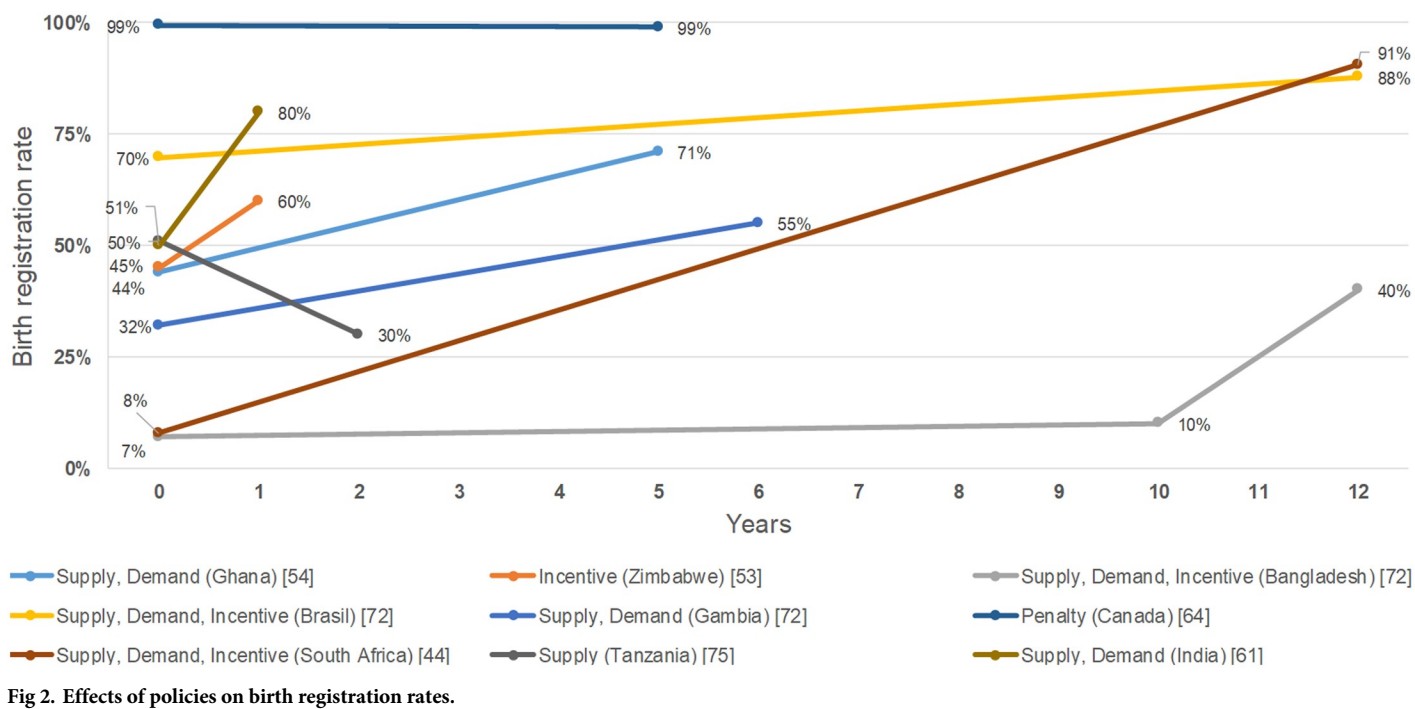

**Fig 2. Effects of policies on birth registration rates.**

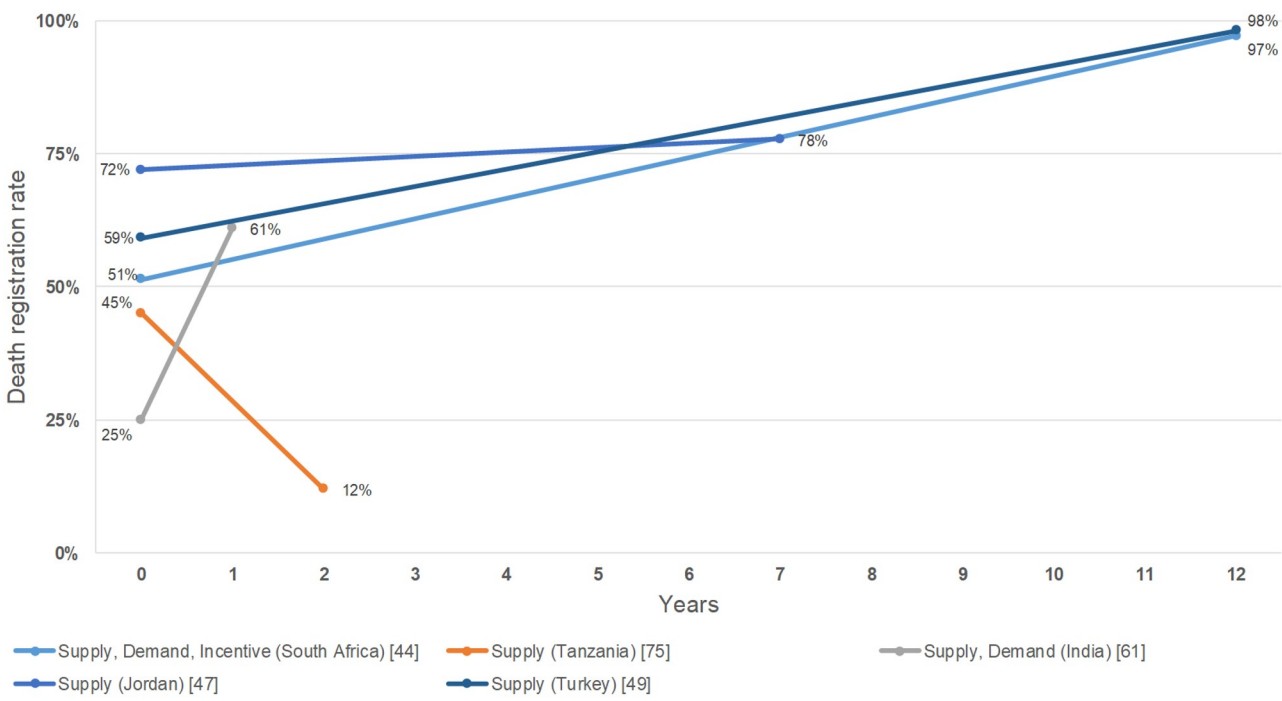

**Fig 3. Effects of policies on death registration rates.**

a subnational scale may face different operational issues and a different direction of effect when implemented nationally. One example of this is use of community-based registration services [47,58,64]. This type of policy would be useful to fill a geographic void in rural settings but may prove redundant in many urban settings. Furthermore, in areas with poor telecommunication platforms and connectivity, mobile registration fills a void; however, in facilities that have computers with connectivity, it may prove redundant [44].

Although most articles reported effects on registration rates, registration rates should be contextualised as part of CRVS business processes [78]. The business process for death typically encompasses five steps: (1) the death itself, (2) notification of the death to a designated site, (3) registration of the death by the registration authority, (4) assigning the death a cause of death, and (5) including the causes of death in the national vital statistics system. The order of the steps is different for health facility deaths, for which the notification typically includes the cause of death. The business process for birth registration encompasses the first three steps and includes the issuance of the birth certificate. Future policy evaluations should report effects by steps in the business process. This may help countries understand policies' effects more comprehensively, identify bottlenecks in the registration process, and work towards improving birth and death registration for their context. For example, Kabadi and colleagues reported that both birth and death registration rates declined after implementing a supply policy in Tanzania [76]. However, closer inspection of the data reveals that birth notifications improved 2-fold, while death notifications improved 5-fold with their intervention. The challenge in Tanzania appeared to be translating these increased notifications to registered events.

Universal access to birth and death registration services is important for clients to register vital events. Supply policy interventions consistently increased birth and death registration rates. The one article that showed a negative direction in the registration of births and deaths from Tanzania, discussed above, found increased birth and death notifications that did not

**Table 2. Operational considerations.**

| Policy | Operational consideration | Country |
|---|---|---|
| Supply | Telecommunication and/or electricity connectivity requirements can hinder information flow in some parts of the country. | Albania [75], Liberia [61], Malawi [74], Peru [56], Tanzania [76] |
| Supply | Telecommunication and physical infrastructure development in parallel with policy implementation facilitated successful rollout. | Ghana [64], Republic of South Africa [45] |
| Supply | Decentralisation to community health workers may be insufficient for large geographic areas that are sparsely populated. | Ethiopia [58], Tanzania [76] |
| Supply | Community-based registration fills a void in settings where births and/or deaths occur outside of a facility, but can be costly. | China [66], Ethiopia [47], Malawi [47], Mali [47] |
| Supply | Paper-based registration systems were not effective and efficient at scale. | India [62], Malawi [52] |
| Supply | Integration of birth and/or death registration in existing services (e.g., health facilities or burial sites) was low cost, required minimal additional human or IT infrastructure, and reduced indirect costs for clients. | Ghana [55], India [73], Papua New Guinea [68], Switzerland [51], US [49] |
| Supply | Partnerships with local academic institutions spurred formative research that helped effectively design policies. | Republic of South Africa [12,60] |
| Supply | Despite the development of strong national legislation, there were challenges due to differences in implementation and enforcement of legislation at the subnational level (e.g., some regions still charged registration fees when they were not required by law). | Brazil [73], The Gambia [73], India [12], Indonesia [46] |
| Supply | Legislation requiring registration, or requiring registration for government services, is insufficient without increasing accessibility and decreasing direct and indirect costs associated with registration. | The Gambia [73], Republic of South Africa [45] |
| Supply and Demand | Identifying the appropriate government cadre to take on new responsibilities, and ensuring appropriate remuneration, helps ensure long-term sustainability of policy implementation. | Ethiopia [47,58], Ghana [64], India [59], Malawi [47], Mali [47], Mongolia [57], Tanzania [76] |
| Supply and Demand | Human resource training and oversight, through government mechanisms or in partnership with local universities, is required to successfully implement new policies. | Albania [75], Botswana [72], Brazil [71], Ghana [64], India [59,73], Malawi [52], Peru [56] |
| Incentive | Requiring a national ID for government services (e.g., education, social protection, voting, etc.) and subsequently linking birth registration to national ID provision improved human resource availability, physical infrastructure accessibility, and client uptake of birth registration. | Botswana [72], Republic of South Africa [45], Thailand [77] |
| Supply, Demand, and Incentive | Collaboration between government, nongovernmental organisations, bilateral partners, and multilateral partners increased resources and approaches to improve registration. Examples included registration campaigns to improve geographic gaps in coverage with multilateral partners and educational campaigns with community and grassroots organisations. | Botswana [72], Brazil [71], India [62], Republic of South Africa [12] |
| Supply, Demand, and Incentive | While central government coordination was helpful because it allowed multiple ministries to guide implementation and enforcement in Southern Africa, autonomy from government allowed technical issues surrounding birth and death registration to remain independent of changes in political priorities in Peru. | Botswana [72], Peru [69], Republic of South Africa [12,60] |
| Supply, Demand, and Incentive | Combination policy approaches were valuable because they allowed multiple barriers, such as accessibility and awareness, to be addressed simultaneously. | Peru [12], India [62], Bangladesh [71] Botswana [72], Brazil [71,73] |

**Abbreviations**: ID, identification card; IT, information technology

translate into increased registration rates [76]. The policy quantitative and operational considerations synthesised should be considered based on the health system, governance, and sociocultural context of contributing articles. For example, fragile and conflict settings may need to consider a supply policy with community-based registration services, whilst settings with a

high proportion of births and deaths occurring in health facilities would likely benefit from integration of registration services into health facilities. Moreover, the role of good governance should not be discounted [79]. Despite enabling legislation in several countries, enforcement challenges at the subnational levels led to obstacles for clients [12,46,73]. Across operational considerations, caution should be exercised in interpreting the findings due to the small sample size of articles with qualitative data.

Information, education, communication, and effective advocacy through demand-side policies can help strengthen implementation of functional CRVS systems by social and behavioural change. However, this approach may have limited utility without functional CRVS systems in place. Most articles that reported results of demand policies did so for policies that coupled demand along with other approaches. For example, 15 articles reported on a demand policy coupled with at least supply [12,45,55,58,60,62,64,69,70,72,73]. The single article that reported on a demand-only policy did so in a context, Mongolia, where birth and death registration rates already exceeded 90% [57]. Therefore, the challenge in Mongolia was reaching the remaining few rather than increasing wide-scale access. This underscores the need to interpret the findings based on the health system, governance, and sociocultural context of contributing articles.

Countries took different approaches to utilising incentives for vital event registration, most of which included provision of a national identification card (ID) or certificate that was required for public service(s). Thailand and Botswana integrated national ID provision, which is required for a wide range of government services, with birth registration [72,77]. Ensuring a life cycle of identity, starting with ensuring that every child is registered and issued a birth certificate and ending with recording the cause of death and providing the family with a death certificate, will maximise the full spectrum of rights and data generated by CRVS systems. For example, infants and young children are able to access life-saving medical interventions, including vaccines and access to early childhood education and nutrition services [80,81]. For children and adolescents, a birth certificate is the first line of defense to be protected against child marriage, labour, and recruitment into armed forces. A birth certificate can also provide the legal identity needed for educational exams and to access higher education and the formal job market. For adults, they enable financial inclusion, social assistance, insurance, inheritance, and land rights. In Brazil, a birth certificate was required to enroll into the Bolsa Familia programme, in which a cash transfer was conditioned upon school enrollment [63]. In South Africa, birth certificates were required for child support grants and school enrollment [12,45,60]. Botswana and Bangladesh also require birth certificates for school enrollment [72,73]. Death certificates provide the evidence needed to access inheritance and land rights, often combined with marriage certificates to prove the legal family ties that establish rights to inheritance [10]. This is especially important for women whose access to financial services and property are often through their husband. Death certificate use was more rare; Botswana reported requiring a death certificate for body burial, while Bangladesh reported requiring a death certificate to pass on inheritance [72,73]. There were no articles evaluating direct financial incentives for registering births or deaths.

Due to limited data, it was not possible to conduct subanalyses. If data permitted, analysing effects by the country CRVS functional level would help inform policy categories that are most relevant based on their current CRVS system functional level. For example, any emerging policy to record births or deaths is unlikely to boost the registration rate if the key elements such as legal framework and national ID are not supported by a current policy. Further research is needed to validate these hypotheses. Nonetheless, combination approaches tailored to country context uniformly and consistently improved birth and death registration rates. Fifteen of the sixteen articles reporting on combination approaches included at least a supply and demand

element, while six included a supply, demand, and incentive element. South Africa reported positive experiences collaborating with local universities in order to research how best to design supply policy interventions [12,60].

The exclusion of individuals from registering a birth (establishing their legal identity) or obtaining a birth certificate (proof of legal identity) is often an unintended consequence of policy, rather than a deliberate effort by the state to exclude them. For example, laws (1) requiring documentary prerequisites (e.g. ID, birth/marriage certificates), (2) with paternity requirements (i.e. name or presence of the father), and/or (3) that impose fines/fees to discourage late registration may be designed with the intention of legitimising registration documents but can deter or legally even prevent women and the most vulnerable members of society from being included in civil registration systems. This systematic review did not identify evaluations of these types of policy changes.

There may be limitations with this systematic review. First, there is possibility of publication bias because many countries, or regions such as Europe and the Eastern Mediterranean, may not have documented policy interventions. Other countries may have documented changes in registration rates without explaining contributing policy shifts [82]. Finally, policies that did not lead to positive effects may not have been documented at all by their investigators. Knowledge of what does not work is also critical for effective CRVS strengthening in countries. Moreover, given that there was no systematic way to search the grey literature, our grey literature search may have been limited in its sensitivity in the global stakeholders contacted. Including additional global stakeholders, such as the European Union, Plan International, United Nations Population Fund, and the United Nations High Commissioner for Refugees, as well as regional stakeholders such as regional development banks, regional economic commissions of the United Nations, and regional technical assistance institutions, may further expand reach. We may have also missed relevant literature that predated our inclusion criteria [83]. Because numerators and denominators for registration were not reported, meta-analysis was not possible. There were several methodological limitations of the studies included. First, given the limited amount of methodological detail provided by eligible articles, it was not possible to ascertain the quality of studies as was planned in the protocol. Second, most eligible studies were included on the basis of their qualitative data and did not have a comparator group to quantify policy effects. Finally, both the design and analytical approach of included quantitative articles limited inferences on causality. This underscores the need to carefully design, monitor, evaluate, and document future CRVS policies.

We identified several gaps in this systematic review. First, more information is needed on financial and human resource requirements for different policies. This information was rarely reported and is critical information required for decision-makers considering national or subnational implementation. Second, many of the articles had a short duration; continued monitoring is needed to understand effects of policies over time. Third, the articles did not provide detail on how migratory populations were registered for births and deaths and whether the findings presented were generalisable to them. Fourth, this systematic review focused on birth and death registration data directly collected from CRVS systems; including whether the policies had effects on additional health outcomes merits further research.

In conclusion, to our knowledge this systematic review provides the first comprehensive compilation of national policies to strengthen birth and death registration. Effective and enforced policies will play a critical role in improving birth and death registration globally. Countries will need to understand their barriers to identify which policy approaches are most appropriate to their context. Further research using a more systematic approach to intervention design and evaluation is needed to improve the knowledge of effects, resource requirements, and acceptability of policies for strengthening CRVS systems.

**Disclaimer**: The statements in this article are those of the authors and do not necessarily represent the official position of their organisations or funding agencies.

## Supporting information

**S1 Text. Protocol.**
(PDF)

**S2 Text. PRISMA checklist.**
(PDF)

**S3 Text. Systematic review search strategy.**
(PDF)

**S1 Table. Quantitative results of eligible articles.**
(PDF)

**S2 Table. Qualitative results of eligible articles.**
(PDF)

## Author Contributions

**Conceptualization:** Amitabh Bipin Suthar.

**Data curation:** Amitabh Bipin Suthar, Aleya Khalifa, Sherry Yin, Kristen Wenz, Doris Ma Fat, Samuel Lantei Mills, Erin Nichols, Carla AbouZahr, Srdjan Mrkic.

**Formal analysis:** Amitabh Bipin Suthar, Aleya Khalifa, Sherry Yin.

**Funding acquisition:** Amitabh Bipin Suthar.

**Methodology:** Amitabh Bipin Suthar, Aleya Khalifa, Sherry Yin, Kristen Wenz, Doris Ma Fat, Samuel Lantei Mills, Erin Nichols, Carla AbouZahr, Srdjan Mrkic.

**Supervision:** Amitabh Bipin Suthar.

**Validation:** Amitabh Bipin Suthar, Aleya Khalifa, Sherry Yin.

**Visualization:** Amitabh Bipin Suthar.

**Writing – original draft:** Amitabh Bipin Suthar.

**Writing – review & editing:** Amitabh Bipin Suthar, Aleya Khalifa, Sherry Yin, Kristen Wenz, Doris Ma Fat, Samuel Lantei Mills, Erin Nichols, Carla AbouZahr, Srdjan Mrkic.

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
