## [Decision Letter · Decision Letter 0]

23 Jul 2019

Dear Dr. Suthar,

Thank you very much for submitting your manuscript "Towards universal civil registration and vital statistics systems: a systematic review and synthesis of policies to improve birth and death registration" (PMEDICINE-D-19-01902) for consideration at PLOS Medicine. 

Your paper was evaluated by a senior editor and discussed among all the editors here. It was also discussed with an academic editor with relevant expertise, and sent to three independent reviewers, including a statistical reviewer. The reviews are appended at the bottom of this email and any accompanying reviewer attachments can be seen via the link below:

[LINK]

In light of these reviews, I am afraid that we will not be able to accept the manuscript for publication in the journal in its current form, but we would like to consider a revised version that addresses the reviewers' and editors' comments. Obviously we cannot make any decision about publication until we have seen the revised manuscript and your response, and we plan to seek re-review by one or more of the reviewers. 

We expect to receive your revised manuscript by Aug 13 2019 11:59PM. Please email us (plosmedicine@plos.org) if you have any questions or concerns.

We look forward to receiving your revised manuscript. 

Sincerely,

Thomas McBride, PhD

Senior Editor 

PLOS Medicine

plosmedicine.org

1- Thank you for providing your PRISMA statement. Please replace the page numbers with paragraph numbers per section (e.g. "Methods, paragraph 1"), since the page numbers of the final published paper may be different from the page numbers in the current manuscript.

2- Please edit the first section of the title to be less aspirational and more descriptive of the current study.

3- Please include the full dates of your search in the Abstract.

4- In the last sentence of the Abstract Methods and Findings section, please describe the main limitation(s) of the study's methodology.

5- At this stage, we ask that you include a short, non-technical Author Summary of your research to make findings accessible to a wide audience that includes both scientists and non-scientists. The Author Summary should immediately follow the Abstract in your revised manuscript. This text is subject to editorial change and should be distinct from the scientific abstract. Please see our author guidelines for more information: https://journals.plos.org/plosmedicine/s/revising-your-manuscript#loc-author-summary

6- The Abstract Conclusions, as well as the end of the discussion, should be more qualified (“Our systematic review suggests…” or similar).

7- I believe the data statement should point to the primary articles, not the SI files. 

8- Line 345, please add “to our knowledge” to qualify your assertion of primacy.

Comments from the reviewers:

Reviewer #1: Alex McConnachie, Statistical Review

Suther et al provides a report on published evidence of the impact of policy initiatives to improve birth and death registration; this review looks at the use of statistics in the paper.

There is not much for me to review. The figures are quite good. The lack of meta-analysis is justified. My one thought in reading the paper is whether there might be some publication bias, if those implementing unsuccessful policies are reluctant to publicise their findings, but the authors acknowledge this as a limitation of their study. Overall, I found the paper to be interesting to read, and well written.

Reviewer #2: Overall a well written and interesting paper which fills a gap in current literature. A few minor points regarding specific language used below: 

Line 82-83: The sentence indicates that household surveys are mainly used to monitor registration levels. It should be highlighted that in the absence of CRVS data, household surveys are used not only for registration completeness but also for key population indicators. 

line 85-89, it is not only about health planning but planning more broadly

line 110-110: the references about slow progress are a bit outdated (most recent 2014) there has been a lot of progress since then (particularly in South Asia). Also, is an improvement from 58-63 per cent in 15 years slow progress?

line 245-246: registration coverage or registration completeness?

Reviewer #3: 

The topic of the impact of national CRVS implementation policies on outcome on birth and death registration as well as timeliness of registration is an important one as the Lancet 2007 (Who Counts) and 2015 (Counting Births and Deaths) series indicate. In the 2015 series, Phillips et al used modelling to conclude that improved CRVS performance coincided with improved health outcomes.

This MA and SR reviews the evidence on national policy interventions to improve birth and death registration. It does not attempt to correlate policy interventions with improved health outcomes.

The methodology is solid; however, the policies were heterogeneous, the sample size small, and many regions of the world were not represented in the available literature. While this is not the fault of the authors, it does limit the conclusions.

Given these limitations, I think this manuscript would better placed in a more specialized journal. 

Specific queries for the authors:

Making casual links between the policies and changes in birth or death rates would be challenging as policy effects must be taken into account with other country specific contexts such as political and economic instability, especially in fragile states. More context understanding would be important and strengthen this associations (as mentioned by the authors). 

The analysis of the operational considerations was very interesting. However, conclusions that can be drawn might be limited to the small sample size. 

The applicability of one policy to another country may be limited given the differences in contexts. For example, the conclusion that a demand only policy might not be effective was based on an n=1. 1 reported this (Mongolia) where 90% coverage was already achieved.

Methods:

24/33 studies had no comparator group which means the effect of supply, demand, and intervention policies on birth registration was determine from small sample size (n=9). Pls address this in discussion

Similarly Figure 4 only has 5 studies reporting policy effect on death registration. What conclusions can be made from this?

How can the authors conclude a new national CRVS policy has a casual effect and how can one draw conclusions from the heterogeneous types of policy intervention and the varying country contexts? There will be so much noise from whatever else is happening in the country at the time that the actual effect of the new policy is difficult to determine.

This methodology does not capture migratory populations

What about costing of vital records systems? The authors mention "financial"

Europe not represented. Is this bc they are already have robust systems?

No comments on this in paper. Even historicaly? 

Is there precedent that CRVS national policies impact health outcomes (other than modelling studies)?

Can Taiwan really be considered a territory of China? (Lu, 2002)?

Past-present tense issues

The Village Executive Officer can access this application and view trends in notifications, and even see when a notification was made but no certificate was acquired. 

One of the sections is termed, "Impact on birth and death registration rates  

Can you make such a strong statement as to say a specific policy had a negative or positive change on birth or death registration rates? This suggests casuality. Wouldn't association be a more accurate term?

[LINK]

---

## [Editor Report · Decision Letter 1]

19 Aug 2019

Dear Dr. Suthar,

Thank you very much for re-submitting your manuscript "Strengthening civil registration and vital statistics systems: a systematic review and synthesis of policies to improve birth and death registration" (PMEDICINE-D-19-01902R1) for review by PLOS Medicine.

I have discussed the paper with my colleagues and the academic editor. I am pleased to say that provided the remaining editorial and production issues are dealt with we are planning to accept the paper for publication in the journal.

[LINK]

Please also check the guidelines for revised papers at http://journals.plos.org/plosmedicine/s/revising-your-manuscript for any that apply to your paper.

We look forward to receiving the revised manuscript by Aug 26 2019 11:59PM. 

Sincerely,

Thomas McBride, PhD

Senior Editor 

PLOS Medicine

plosmedicine.org

Requests from Editors:

1- Thank you for editing the Title. I still think it could be more descriptive though. Perhaps: “Evaluation of approaches to strengthen civil registration and vital statistics systems: a systematic review and synthesis of policies in 25 countries”

2- Thank you for including an Author Summary. Some of the bullet points are a bit lengthy (eg, point 3 of the “What Did the Researchers Do and Find?” section), please cut a bit or split into multiple bullets (but not sub-points).

3- The limitation that most studies did not have a comparator group could be mentioned in the Discussion, along with the other methodological limitations of the studies included.

4- Discussion limitations could also note the geographic limitations, eg very few European countries represented.

5- Please include your response regarding Taiwan inthe main text.

6- Line 253: "Association with birth and death registration rates"?

---

## [Editor Report · Decision Letter 2]

30 Aug 2019

Dear Dr. Suthar, 

On behalf of my colleagues and the academic editor, Dr. Margaret Kruk, I am delighted to inform you that your manuscript entitled "Evaluation of approaches to strengthen civil registration and vital statistics systems: a systematic review and synthesis of policies in 25 countries" (PMEDICINE-D-19-01902R2) has been accepted for publication in PLOS Medicine. 

PRODUCTION PROCESS

PRESS

PROFILE INFORMATION

Thank you again for submitting the manuscript to PLOS Medicine. We look forward to publishing it. 

Best wishes, 

Thomas McBride, PhD

Senior Editor 

PLOS Medicine

plosmedicine.org